# Transcriptomic and Behavioral Studies of Small Yellow Croaker (*Larimichthys*
*polyactis*) in Response to Noise Exposure

**DOI:** 10.3390/ani12162061

**Published:** 2022-08-13

**Authors:** Xuguang Zhang, Jun Zhou, Wengang Xu, Wei Zhan, Huafeng Zou, Jun Lin

**Affiliations:** 1Engineering Technology Research Center of Marine Ranching, College of Marine Ecology and Environment, Shanghai Ocean University, Shanghai 201306, China; 2School of Ocean, Yantai University, Yantai 264005, China; 3Institute of Hydrobiology, Zhejiang Academy of Agricultural Sciences, Hangzhou 310021, China; 4National Demonstration Center for Experimental Fisheries Science Education, Shanghai Ocean University, Shanghai 201306, China; 5The Key Laboratory of Exploration and Utilization of Aquatic Genetic Resources, Ministry of Education, Shanghai Ocean University, Shanghai 201306, China

**Keywords:** *Larimichthys polyactis*, noise exposure, transcriptomic analysis, extracellular matrix, locomotor behavior

## Abstract

**Simple Summary:**

Ocean noise pollution from marine traffic may affect the behavioral, ecological and biochemical parameters of marine fish species. Most studies have focused on behavioral changes and hearing damage in fishes, but the molecular mechanism of noise exposure in the impairment of the brain has rarely been reported. In this study, using small yellow croaker (*L. polyactis*) as a model, we used RNA-seq methods to characterize differently expressed genes between the control group and the noise exposure group. GO and KEGG pathway analysis found that synaptic transmission, neurotransmitter transport, endocytosis procession, cell adhesion molecules and the extracellular matrix receptor interaction pathway were over-represented in the DEGs. In addition, behavioral studies revealed that *L. polyactis* kept motionless on the surface of the water and lost the ability to keep their balance after noise exposure. Collectively, our results indicate that exposure to noise stressors contributes to neurological dysfunction in the brain and impaired locomotor ability in *L. polyactis*.

**Abstract:**

Noise has the potential to induce physiological stress in marine fishes, which may lead to all sorts of ecological consequences. In the current study, we used the RNA-sequencing (RNA-seq) method to sequence the whole transcriptome of the brain in small yellow croaker (*Larimichthys polyactis*). The animals were exposed to a mix of noises produced by different types of boat played back in a tank, then the brain tissues were collected after the fish had been exposed to a 120 dB noise for 0.5 h. In total, 762 differently expressed genes (DEGs) between the two groups were identified, including 157 up regulated and 605 down regulated genes in the noise exposure group compared with the control group. Gene ontology (GO) enrichment analysis indicated that the most up regulated gene categories included synaptic membranes, receptor-mediated endocytosis and the neurotransmitter secretion process. The Kyoto Encyclopedia of Genes and Genomes (KEGG) pathways found that endocytosis, cell adhesion molecules and the extracellular matrix (ECM) receptor interaction pathway were over-represented. Specifically, ECM-related genes, including *lamin2*, *lamin3*, *lamin4*, *coll1a2*, *coll5a1* and *col4a5* were down regulated in the noise exposure group, implying the impaired composition of the ECM. In addition, the behavioral experiment revealed that *L. polyactis* exhibited avoidance behaviors to run away from the noise source at the beginning of the noise exposure period. At the end of the noise exposure period, *L. polyactis* kept motionless on the surface of the water and lost the ability to keep their balance. Taken together, our results indicate that exposure to noise stress contributes to neurological dysfunction in the brain and impaired locomotor ability in *L. polyactis*.

## 1. Introduction

The amount of anthropogenic noise in the marine environment caused by commercial shipping, hydroelectric power plants and drilling is increasing [1]. There is growing concern about the effects of this anthropogenic noise on aquatic life, since many fishes depend on sound to communicate, detect prey or predators, and respond to the world around them. Noise pollution may affect the behavioral, ecological and biochemical parameters of marine fish [2,3].

Anthropogenic noise sources, which are mainly produced by boats emitting noise (typically <1 kHz), occupy the same frequency bands that many marine species use to communicate [4]. The highest intensities usually fall within frequencies ranging from 0.1 to 1 kHz [5]. This frequency range has been shown to be a potential threat to fish species, because most audiograms of marine fish species show that their greatest sensitivity to noise falls within this range [6]. More studies have demonstrated that anthropogenic noise may affect the hearing thresholds, communication, movement patterns, foraging and biochemical parameters of fish, and even the fitness of ecological populations [7]. For example, recent studies found that fish escaped and swam faster [8], but suffered more mortality by predation with exposure to boat noise [9].

In the sea environment, noise pollution may pose adverse effects related to orientation toward prey, predators and conspecifics in their environment. In aquaculture systems, cultured fish could also be exposed to noise from the aerator, water pumps and filtration systems [10]. For example, Huang et al. observed that largemouth bass had significantly lower weight gain after exposure to underwater noise, accompanied by a higher degree of oxidative stress in the liver tissues [11]. In studies on rats, it was found that noise exposure could induce structural and functional injuries of the brain, which resulted in impairments in memory and deficits in motor coordination [12].

The small yellow croaker (*Larimichthys polyactis*, *L. polyactis*), a demersal fish species, is widely distributed in the Bohai sea, East Sea and Yellow Sea of China [13]. Due to its high economic and nutritional value, *L. polyactis* has become an economically important marine fish. To date, studies of noise pollution on aquatic animals have mainly focused on behavior and hearing damage, and the detailed mechanisms underlying the intracellular responses to noise exposure in the brain are lacking. In the present study, we applied RNA-seq to sequence the whole transcriptome of the brain tissues of *L. polyactis* to identify transcriptome changes after noise exposure.

## 2. Materials and Methods

### 2.1. Animal

Wild *L. polyactis* fish with body weights of approximately 50–70 g were provided by the Marine Fisheries Research Institute of Zhejiang (Hangzhou, China). Fish were transported to Shanghai Ocean University and cultured at 14 °C for 1 week in a 1 m^3^ tank for acclimation before the beginning of the experiments. All animal procedures were approved by the Animal Ethics Committee of Shanghai Ocean University (series number: SHOU-DW-2021-022).

### 2.2. Sound Exposure and Behavior Experiment

The *L. polyactis* fish were exposed to boat noise with a mean sound pressure level (SPLrms) of 120 dB re 1 µPa continuously for 30 min in soundproof room (Shino Acoustics, Shanghai, China). The noise from a wooden fishing boat with a 22-horsepower outboard engine was recorded in the waters of a suspended mussel farm. The boat noise was played back through a portable computer via Audacity 2.4. After passing through the power amplifier (Sansui S2-350, Guangzhou, China), the signal was fed to a UW30 underwater speaker with a frequency response of 100–10 kHz (Figure 1). The speaker was set facing upward at the bottom of an acrylic cylindrical tank with a diameter of 40 cm and filled to 45 cm depth with seawater, without the speaker touching the side of the tank. During exposure, a floating nylon cage (20 cm × 20 cm × 20 cm) 5 cm above the speaker was used to hold the fish. To monitor ambient tank sounds and noise playback during the experiments, a Brüel & Kjær hydrophone (type 8013, with a sensitivity of −212 dB re 1 µPa and a frequency range of 0.1 Hz to 20 kHz) was deployed into the cage to measure the recorded signal without fish present. The signal from the hydrophone was connected to a Brüel & Kjær condition amplifier (type 2692), then fed to the Brüel & Kjær 3050 input module, saved and analyzed by BK Cnnect (version 2009.1) software. The root mean square sound pressure level (SPLrms) of the sound recordings was computed using BK Connect (Fast Fourier Transform lines 1600, Hanning window with 66.7% overlap between 10 and 2000 Hz). Since most fish species are sensitive to frequencies between 100 and 1000 Hz [14], the frequency range of 10–2000 Hz was used in this study. During the process of noise exposure, the locomotor behavior of the fish in the tank was recorded using a video camera.

### 2.3. Total RNA Isolation and Illumina Sequencing

After the experiment of noise exposure was finished, the fish were removed from the tank and sacrificed. The fish in the control group were put in the same tank but were not exposed to the noise stress. The tissues of the brain (control, *n* = 3; noise, *n* = 3) were quickly dissected and flash-frozen in liquid nitrogen. Total RNA was isolated using Trizol reagent (Invitrogen, Carlsbad, CA, USA) according to the manufacturer’s instructions. For RNA-seq, total RNA was subjected to poly(A) enrichment using Oligod(T)x25 Magnetic Beads. RNA-seq libraries were prepared using the NEBNext mRNA Library Prep Reagent Set (NEB; lpswich, MA, USA). Sequencing was conducted with the Illumina HiSeq 1500 platform (Oebiotech, Shanghai, China). All the raw sequence data of RNA-seq have been deposited in the National Center for Biotechnology Information (NCBI) Sequence Read Archive (SRA) under BioProject accession number PRJNA862477.

### 2.4. Alignment of Transcriptomic Data

After trimming low-quality bases from the 5′ and 3′ ends of the remaining reads, the resulting high-quality reads were mapped onto the *L. polyactis* reference genome [15]. An index of the reference genome was built, and clean reads were aligned to the reference genome of *L. polyactis* (Genbank assembly accession: GCA_018985215.1) using hisat2. The HTSeq-count was used to generate the total number of reads for each gene [16]. For differential gene expression testing, gene-level counts were analyzed using the R bioconductor package edgeR [17]. Differentially expressed genes (DEGs) were considered statistically significant if the false discovery rate (FDR) was 0.05 and the absolute value of log_2_ (fold change) ≥ 1.

### 2.5. GO and KEGG Pathway Enrichment Analysis

Gene Ontologies (GO) were obtained from the mapped translated protein sequences and homology searches against Swissprot using BLASTX in DIAMOND with an E-value cut-off of 1 × 10^−5^ to retrieve the ID mapping [18]. In order to identify characteristic changes associated with noise exposure, GO term enrichment analysis was performed. Kyoto Encyclopedia of Genes and Genomes (KEGG) annotations were obtained using eggNOG mapper [19]. The GO functional enrichment and KEGG pathways analyses were performed using the R package clusterProfiler 4.0 [20].

### 2.6. Quantitative Real-Time PCR

Reverse transcription (RT) reactions were carried out using the Hifair^®^ Ⅱ 1st Strand cDNA Synthesis Kit (Yeasen Biotechnology CO., Ltd., Shanghai, China) following the standard protocol. The qRT-PCR test was conducted by using SYBR Green Premix ExTaq (Takara, Dalian, China) on the ABI 7500system (Applied Biosystems, Carlsbad, CA, USA). The relative mRNA expression of related genes was calculated by reference to β-actin using the 2^−ΔΔCt^ method [21]. The primers were designed using Primer Premier 5.0 software.

## 3. Results

### 3.1. Transcriptome Overview

In this study, 130 million 150 bp paired-end reads were generated for the RNA-seq analysis. The six libraries generated 21.67 M raw reads on average, which resulted in 21.3 clean reads on average and a Q30 value ranging from 91.2% to 93.6%. One sample in the noise group was corrupted and excluded from further analysis. In total, 25,233 assembled transcripts were obtained and mapped to the genome of *L**. polyactis*.

### 3.2. Gene Differential Expression Analysis

We found that 762 genes were differentially expressed between the control group and the noise exposure group. Among these, 157 were up regulated and 605 were downregulated in the noise exposure group compared with the control group (Appendix A). A volcano plot was used to visualize the differential expression of transcripts between the control and noise exposure groups (Figure 2). The red and green circles indicate up and down regulated genes, respectively. In addition, the heatmap of the DEGs provided a visual illustration of the expression differences (Figure 3), revealing a clear pattern that distinguished between the control and noise exposure groups.

### 3.3. Gene Ontology (GO) Analysis of Different Expressed Transcripts between the Two Groups

To obtain more information about the mechanism of how the brain responds to noise stress, Gene Ontology (GO) analysis was performed to elucidate the biological implications of the DEGs in the experiment. Synaptic membrane, postsynaptic membrane, receptor-mediated endocytosis and neurotransmitter secretion in the biological process category were enriched (Figure 4).

### 3.4. KEGG Pathway Enrichment Analysis of DEGs

KEGG pathway enrichment analysis of the DEGs was performed to find critical pathways related to noise exposure stress. As shown in Figure 5, the KEGG pathway enrichment analysis identified that the pathway of endocytosis, ECM-receptor interaction and cell adhesion molecules were affected by noise exposure. Since laminin and collagen genes are closely related to the ECM and cell adhesion molecule pathway, these related genes were analyzed. It was found that almost all the related genes (*coll1a2*, *coll5a1*, *col4a5*, *LOC104926245*, *LOC104931259*, *lamin2*, *lamin3* and *lamin4*) were down regulated in the noise exposure group compared with the control group (Figure 6).

### 3.5. Validation of the RNA-seq Results Using RT–PCR Methods

Real-time reverse transcription polymerase chain reaction (RT–PCR) analysis of the expression of eight genes was used to verify the RNA-seq results. Both methods identified genes with significant changes in their expression levels, both upregulated genes (*nek10*, *ccdc141*, *fkbp6* and *gp1bb*) and downregulated genes (*cops9*, *lama3*, *fat2* and *grid2*). The results of the qRT-PCR analysis were consistent with the RNA-seq results (Figure 7), suggesting the reliability of the RNA-seq results.

### 3.6. Locomotor Behavior of L. polyactis after Noise Exposure

Since our results demonstrated that noise exposure resulted in dysfunction the brain of *L. polyactis*, we tried to evaluate if the transcriptomic change in the brain could be related to behavioral changes induced by noise exposure. The behavior experiment revealed that when *L. polyactis* fish were exposed to noise, they exhibited an obvious avoidance response. At the end of the noise exposure period, *L. polyactis* lay motionless on the surface of the water and lost the ability to keep their balance (Appendix A and Figure 8).

## 4. Discussion

Although environmental noise exposure has been extensively studied, most studies have focused on behavioral changes and hearing damage in fishes and mammals [22]; the molecular mechanism of noise exposure in the brain has rarely been reported. Hence, in this study, we investigated the exposure of *L. polyactis* to noise pollutants to evaluate the effects of noise stress on the brain.

### 4.1. Synaptic Dysfunction

Synapses depend heavily on cell adhesion to provide a physical link between pre- and post-synaptic cells. The adhesion of cells to cells or the extracellular matrix regulates the structure and function of synapses [23]. Synaptic disruption has been recognized as an early neuropathologic event. In this study, GO enrichment analysis indicated that the synaptic membrane, postsynaptic membrane, endocytosis and neurotransmitter secretion were significantly enriched in the cellular component category and biological process category. These results supported the evidence that the noise stress alters the function of synapses [24]. Vesicular inhibitory amino acid transporter (VIAAT) is a synaptic vesicle protein responsible for the vesicular storage of γ-aminobutyrate (GABA) and glycine [25]. Regulating synaptic membrane exocytosis 4 (*rims4*) proteins are important components of the presynaptic machinery for neurotransmitter release, and perform an essential function in neurotransmitter release. In this study, all of these genes were found to be significantly down regulated after noise exposure. In addition, the structural and biochemical changes in the synapses in response to a number of stimuli are referred to as synaptic plasticity [26], which includes alterations in postsynaptic proteins and morphological changes [27]. Morphological changes can cause functional changes and lead to dysfunction in synaptic transmission efficiency [28]. Dysfunctions of the synapse may result in abnormal locomotor behavior, which is consistent with the motionless behavior of *L. polyactis* observed at the end of the noise exposure experiment.

### 4.2. Neurotransmitters and Synaptic Transmission

The neurons communicate via chemical synapses. Neurotransmitters are stored in synaptic vesicles and released by exocytosis upon activation. After these synaptic vesicles have been released, they need to be recycled locally in order to maintain synaptic transmission [29]. Noise stress has also been shown to alter neurotransmitter levels in the brain and reduce dendrite growth, leading to impaired memory and cognition in rats [30,31,32]. Previous studies have found that acute and chronic noise stress results in the release of neurotransmitters such as glutamate, acetylcholine and gamma-aminobutyric acid (GABA) in the brain [33,34,35]. Accordingly, in this study, we found that several genes related to neurotransmitter transport genes, such as *Grid2* and *LOC104931322*, were over-represented in the GO enrichment analysis. *Grid2* is a member of the ionotropic glutamate receptor family of excitatory neurotransmitter receptors. Changes in the expression and post-translational modifications of *Grid* genes can lead to dynamic changes in the neural circuits in normal learning and diseased states [36]. The gene *LOC104931322*, sodium- and chloride-dependent GABA transporter 2, plays a role in the disposition of GABA in the brain. In this study, both *grid2* and *LOC104931322* were significantly down regulated, suggesting that the transmission of glutamate and GABA was altered by noise stress. Synaptic transmission efficiency was reduced, and this reduction led to synaptic dysfunction.

### 4.3. The Composition of the Extracellular Matrix (ECM)

The ECM of the brain is essential for the maintenance of brain function, providing structural support and mediating cell–cell interactions [37]. Laminin is important components of the ECM and participates in neuronal development, survival and regeneration. Laminin is rich in the basement membranes of the endothelial cells of the blood–brain barrier. Degradation of laminin can affect neuronal survival, and loss of the laminin foundation can predispose neurons to excitotoxic death [38]. For example, an injection of excitotoxin into the hippocampus of mice resulted in hippocampal neuronal death, accompanied by the degradation of laminin [39]. Consistent with this idea, several laminin genes, including *lama2*, *lama3* and *lama4* were down regulated in this study (Figure 5), implying that the composition of the ECM in the brain is destroyed after noise exposure.

In the systemic ECM, the most abundant matrix components are the collagen family, which provide structural integrity and contribute to stability and biomechanical properties in most tissues. Collagens play an important and multifunctional role in the peripheral and central nervous systems [40]. In the study of rats, collagen genes including *col1a1*, *col1a2* and *col5a2* in the cochlea were found to be reduced after loud or very loud noise exposure [41]. It is speculated that collagen proteins provide structural support within the cochlea, and the structures are damaged by excess noise and are subsequently degraded. Similarly, a number of transcripts encoding collagens (*col1a**2*, *col**4a**5*, *6a3*, *col**15a1*, *col1**8a1*, *loc104926245* and *LOC104931259*) in the brain of *L. polyactis* were down regulated by noise exposure in this study, suggesting the damage to the function of the brain.

### 4.4. Linkages between Behavioral Responses and Neurotransmitters

In mice, exposure to a 100 dB noise for 2 h caused excitotoxic trauma to the cochlear synapses and triggered excessive release of the neurotransmitter glutamate from the auditory sensory hair cells [42]. This loss of glutamate homeostasis may initiate secondary injury in the brain through the activation of cell death pathways [43]. In this study, *L. polyactis* showed increased locomotor behavior and tended to swim away from the noise source at the beginning of the noise exposure. This was accompanied by a release of glutamate [44], and accumulated glutamate will inevitably destroy the homeostasis of the neurotransmitters and impair the function of the brain. As expected, at the end of the noise exposure period, *L. polyactis* lost their balance and lay flat on the water, implying the loss of the balance neurotransmitters in the brain. A similar result was reported in rats exposed to noise stress of 100 dB: it was found that rats showed impaired motor coordination in the noise exposure group on Days 1 and 15 when compared with the control group. This impaired motor function could be attributed to the altered neurotransmitter levels in both the striatum and cerebellum in the brain [45].

## 5. Conclusions

In summary, using RNA-seq methods, we revealed a global view of gene transcript regulation in noise-exposed *L*. *polyactis*. We identified differentially expressed genes associated with the extracellular matrix and neurotransmitters in the brain of *L. polyactis* following noise exposure. Our findings may contribute to an in-depth elucidation of the molecular mechanisms of neurological dysfunction and impaired locomotor ability in *L. polyactis* after noise-induced stress responses.

## Figures and Tables

**Figure 1 animals-12-02061-f001:**
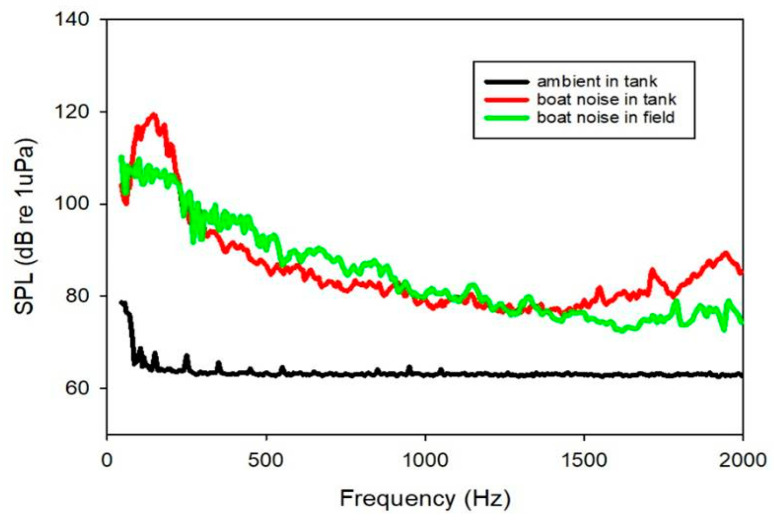
Sound pressure levels (SPL) in the experimental tank during boat noise playback (boat noise in tank) and boat noise recorded in mussel farm water (boat noise in the field) and ambient conditions in the experimental tank without playback (ambient in tank). The main frequency of boat noise in the tank is between 100 and 200 Hz, and owing to the space limitations of the tank, the boat noise in the tank was nearly 10 dB higher than the boat noise in the field.

**Figure 2 animals-12-02061-f002:**
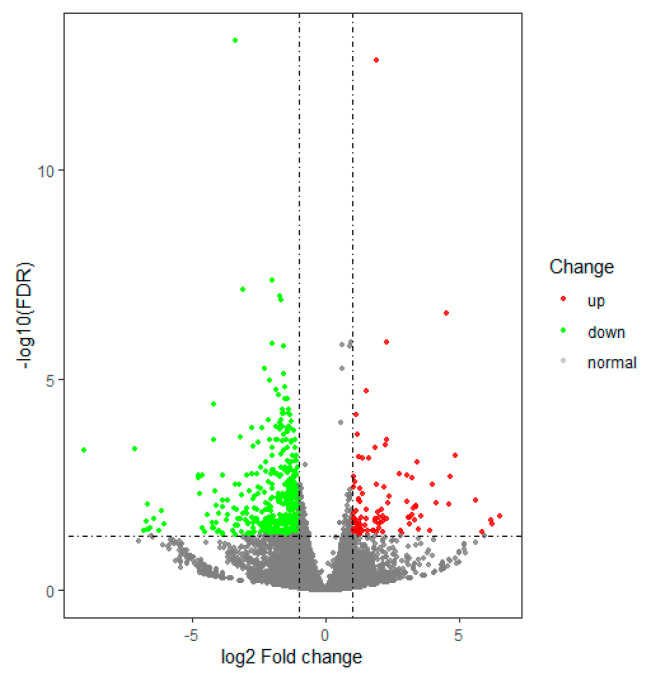
Volcano plot of differentially expressed genes in the control group and noise exposure group. Each dot represents an individual gene. Red dots represent significantly up regulated genes; green dots represent significantly down regulated genes.

**Figure 3 animals-12-02061-f003:**
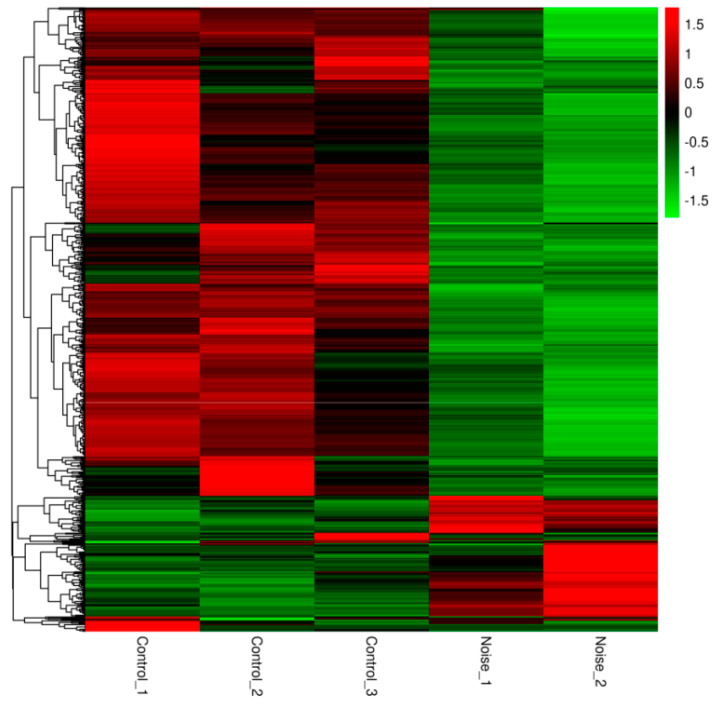
Heat map of DEGs between the control group (Control) and the noise exposure group (Noise). The horizontal axis is the sample name, and the vertical axis is the normalized value of the differential gene in FPKM. Red color in the heat map indicates up regulated genes and green color indicates down regulated genes between groups.

**Figure 4 animals-12-02061-f004:**
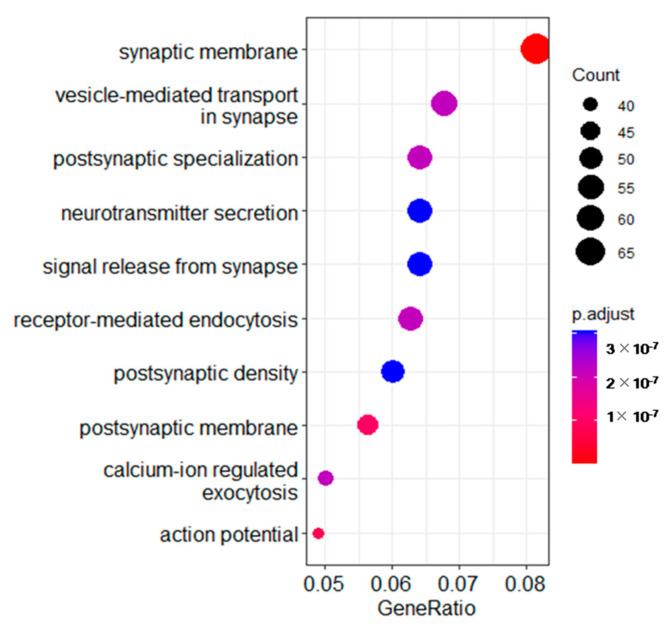
The top 10 most significantly enriched Gene Ontology (GO) terms from the differentially expressed genes. The size of the dots represents the number of genes for different terms. The p.adjust values are indicated by changing colors, moving from red to blue.

**Figure 5 animals-12-02061-f005:**
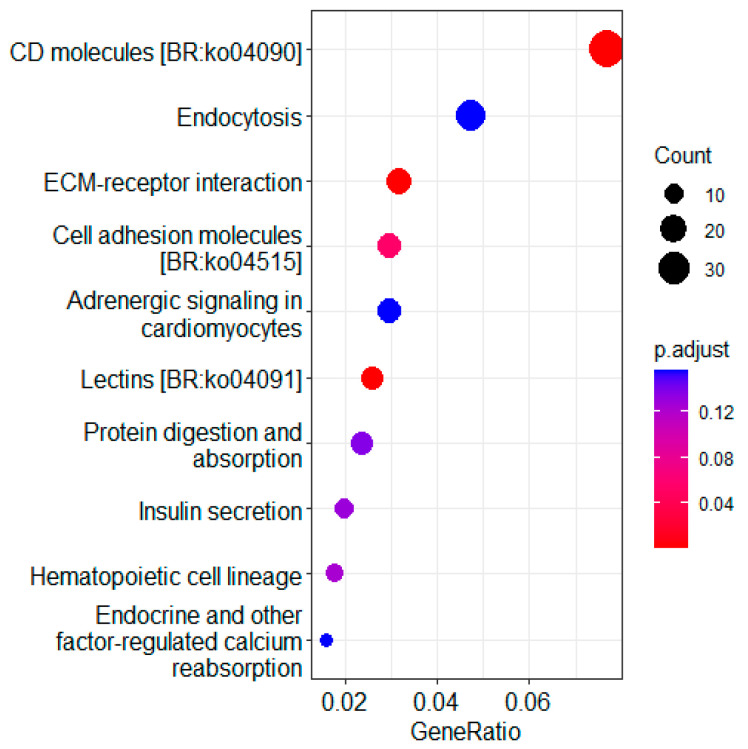
KEGG pathway enrichment analysis of differentially expressed genes between the control and noise exposure groups. The *x*-axis represents the enrichment factor for each of the differentially expressed genes in each pathway. The *y*-axis shows the name of the enriched pathway. The size of each node represents the number of enriched genes in that pathway, and the p.adjust values are indicated by changing colors, moving from red to blue.

**Figure 6 animals-12-02061-f006:**
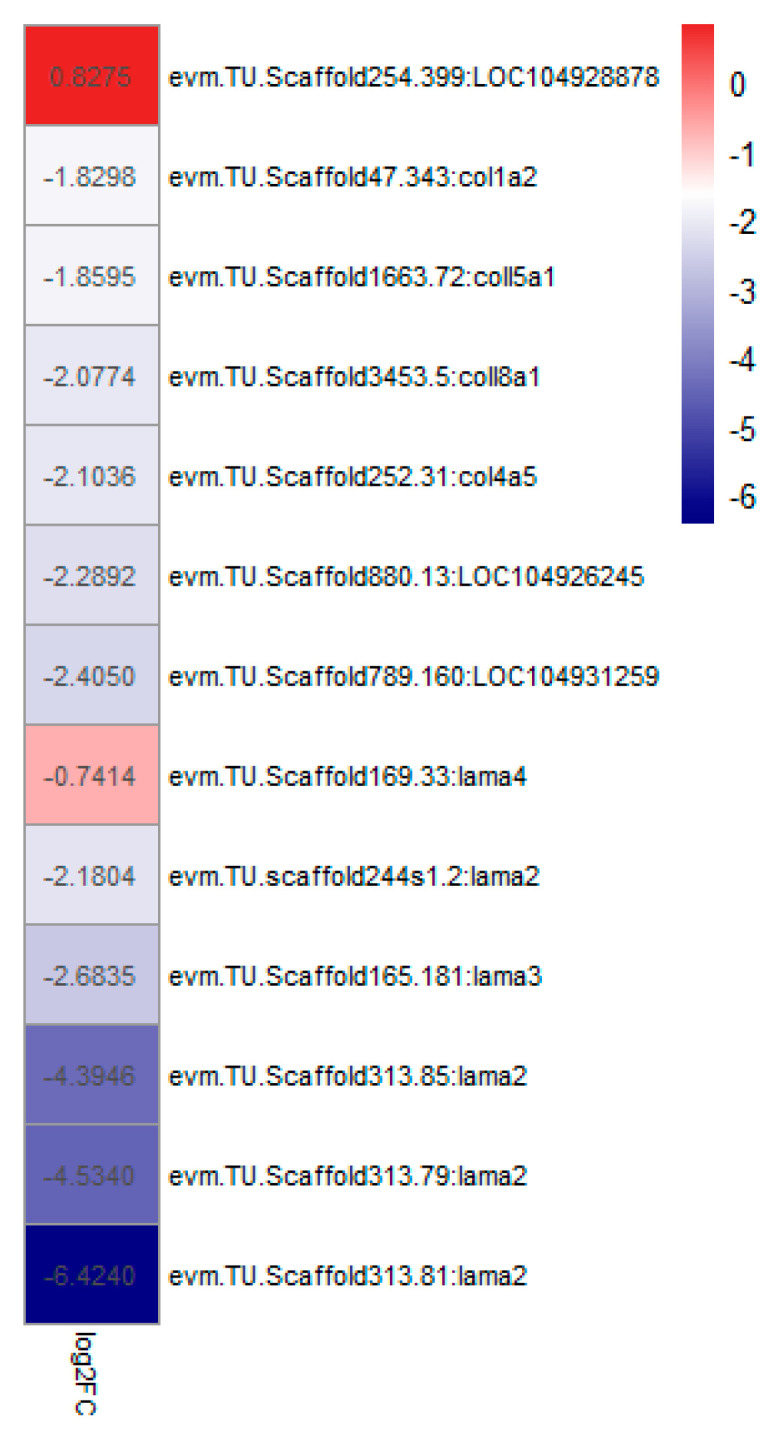
Heatmap showing the change in ECM-related genes (collage and laminin) after noise exposure. Log_2_FC: Fold change of the gene expression level in the exposure group compared with the control group. Red color in the heat map indicates up regulated genes and blue color indicates down regulated genes in the exposure group compared with the control group.

**Figure 7 animals-12-02061-f007:**
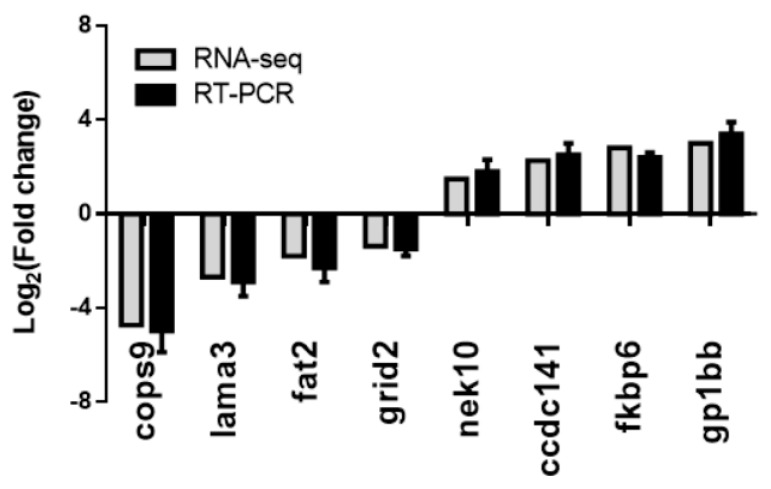
Validation of the RNA-seq results using RT-qPCR methods. The gray columns indicate the RNA-seq results; the black columns indicate the real-time PCR results. The *x*-axis indicates the gene names; the *y-*axis represents the value of log_2_(fold change) of the noise exposure group compared with the control group.

**Figure 8 animals-12-02061-f008:**
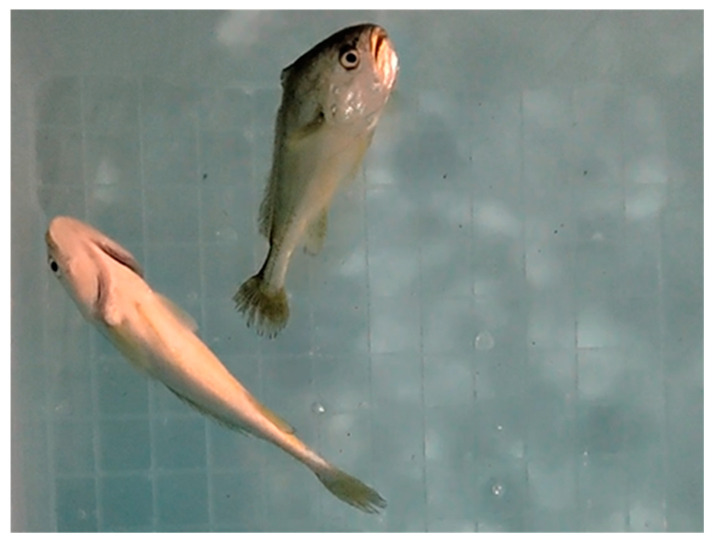
Loss of balance in *L. polyactis* after noise exposure for 0.5 h.

## Data Availability

All the raw sequence data of RNA-seq have been deposited in the National Center for Biotechnology Information (NCBI) Sequence Read Archive (SRA) under BioProject accession numbers PRJNA862477.

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
