# Peer review of "Transcriptomic and Behavioral Studies of Small Yellow Croaker (Larimichthys polyactis) in Response to Noise Exposure"

_animals, 2022, doi:10.3390/ani12162061_

Round 1

Reviewer 1 Report

The manuscript has a very interesting topic, however there are too many inaccuracies and there are unclear and confusing parts.

Introduction

The introduction is too concise. Examples are described in a simple way, other species are not mentioned nor effects on a physiological or ecological level. The authors should implement the introduction by giving more emphasis to the problem related to noise and the effects that this can cause in the aquatic environment

Materials and methods

2.1 : In my opinion one week for acclimation is really short and I believe this could alter the results obtained in subsequent analyzes. So I wonder, have you verified that one week is enough for the species in question or is there a supporting bibliography? Please specify.

Line 65: “Animal Ethics Committee of Shanghai Ocean University”, You should add some authorization code

2.2: How many specimens did you use for the experiment?

2.3: how were the specimens sacrificed? no control group is mentioned. How many replicates?

Results

Line 135: This is your first time talking about control. The experimental groups should be described in the materials and methods. How many specimens per group? 

Line 140: please add “differentially expressed genes” before DEGs. 

What is Figure 1? it is not mentioned in the main text

The quality of Figure 2 needs to be improved and better described. What information should it give if none of the dots represent a gene? Explain this in the main text.

In figure 3 the caption should be improved: indicate CON and EXP. They are not described in the materials and methods and this makes reading difficult and confusing. I would add that, if not described, it is not very significant when compared to figure 2

Section 3.3 There are no results in this section. What should I read in Table 1?

3.5: "Both methods identified genes with significant changes in expression levels: up-regulated genes (mast1, scl12a4, 187 abca2, atp1b3 and scl6a11) and down-regulated genes (sept3, PSAP, grid2, npsa4 and septin-7)" but up and down adjusted with respect to what? where are the gene expression values?

Figure 5 should be moved to the previous section

Author Response

Reviewer #1:

Comments and Suggestions for Authors

The manuscript has a very interesting topic, however there are too many inaccuracies and there are unclear and confusing parts.

Introduction

The introduction is too concise. Examples are described in a simple way, other species are not mentioned nor effects on a physiological or ecological level. The authors should implement the introduction by giving more emphasis to the problem related to noise and the effects that this can cause in the aquatic environment

Reply: Thanks so much for your suggestion.

In this study, little yellow corakde is lived in marine environment, so the noise in sea envirmonent is the main issue we discussed. We agree your opinion, the noise may also affect the growth fish in aquaculture environment, we also added discussions in the MS.

Materials and methods

2.1: In my opinion one week for acclimation is really short and I believe this could alter the results obtained in subsequent analyzes. So I wonder, have you verified that one week is enough for the species in question or is there a supporting bibliography? Please specify.

Reply: Thanks for your suggestion. We agree your opinion that two weeks will be a better acclimation period. However, one week of acclimation is also been in others studies.

Before the experiment of noise exposure, the control group and noise group are keep in the same environment, so the difference of acclamation can be ignored. The study of Fan et al., and Zeng etal., also used one week acclimation time for fish.

1, Fan Z J , Jia Q J , Yao C L . Characterization and expression analysis of Toll-like receptor 2 gene in large yellow croaker, Larimichthys crocea[J]. Fish Shellfish Immunol, 2015, 44(1):129-137.

2, Zeng, Lin, Chang-Wen, et al. Effects of beta-glucan on ROS production and energy metabolism in yellow croaker (Pseudosciaena crocea) under acute hypoxic stress[J]. Fish Physiology and Biochemistry, 2016

Line 65: “Animal Ethics Committee of Shanghai Ocean University”, You should add some authorization code

Reply: Thank for your suggestion. We have added it to the MS. Series number: SHOU-DW-2021-022.

2.2: How many specimens did you use for the experiment?

Reply: In RNA-seq study, control (3 fish) and noise group (2 fish) were used for RNA-seq analysis.

2.3: how were the specimens sacrificed? no control group is mentioned. How many replicates?

Reply: Six fish were sacrificed. In the 2.3 section of MS, we mentioned the control group and noise group. In the RNA-seq study, both the control group and noise group have three samples. However, one sample of noise group was polluted, so two RNA libraries were used for noise group.

Results

Line 135: This is your first time talking about control. The experimental groups should be described in the materials and methods. How many specimens per group? 

 Reply: Thanks for your suggestion.

Experiment group is the noise group. In order to avoid confusing, we used Noise group in the revised MS. The noise group expose to 0.5 h noise stress, while the control group do not expose. In this study, we performed RNA-seq analysis, the control group is 3 fish and noise group has 2 fish.

Line 140: please add “differentially expressed genes” before DEGs. 

Reply: Thanks for your suggestion.

When “differentially expressed genes” was firstly stated in the 2.4 section, we give the “differentially expressed genes” as DEGs.

What is Figure 1? it is not mentioned in the main text

Reply: We added the Fig 1 to the MS. Line

The quality of Figure 2 needs to be improved and better described. What information should it give if none of the dots represent a gene? Explain this in the main text.

Reply: Thanks for your suggestion.

RNA-seq has been reanalyzed. We have revised the figure 2, and added description in the text of MS.

In figure 3 the caption should be improved: indicate CON and EXP. They are not described in the materials and methods and this makes reading difficult and confusing. I would add that, if not described, it is not very significant when compared to figure 2

Reply: Thanks for your question.

In order to avoid the confusing, we have used Noise instead of EXP. We also described it in the figure legend.

In the figure of heatmap (Fig 2 in revised MS), the blue and red colors do not mean the fold change of RNA-data. The data has been changed in logarithmic transformation and converted to z-scores of the rows (genes). This procession of dada make the picture look much better and we can see different patterns picked out by the clustering algorithm.

Section 3.3 There are no results in this section. What should I read in Table 1?

Reply: Thanks for your question.

In section 3.3 we want to show the GO enriched results. In old version, this should be table 2. In the revised MS, we give it to a graph result of GO enriched (Fig 4).

3.5: "Both methods identified genes with significant changes in expression levels: up-regulated genes (mast1, scl12a4, 187 abca2, atp1b3 and scl6a11) and down-regulated genes (sept3, PSAP, grid2, npsa4 and septin-7)" but up and down adjusted with respect to what? where are the gene expression values?

Reply: The figure give the fold change of related gene expression of noise group and control group. The fold change is based on real-time PCR results and RNA-seq results. We give the final results of fold change. The gene expression values of FPKM are also given in Supplement 1.

Figure 5 should be moved to the previous section

Reply: Thanks a lot. It has been moved to the section above.

Reviewer 2 Report

In this manuscript, Zhang et al. investigated the effect of noise exposure on the brain transcriptome. A total of 396 differentially expressed genes were detected, which were enriched in neurological functions. The study is novel and is of interest to the field. However, a technical issue needs to be addressed through reanalysis of the RNA-seq data, before it is suitable for publication at Animals.

Major comment:

The authors claim “we assembled and annotated a de novo transcriptome and compared transcript levels in the control group and noise exposure group. Totally 72421 unigenes with high quality.” This is not correct. In general, fish genomes consist of 20 to 40 thousand protein-coding genes. The 72421 genes detected in this research cannot be high-quality. There must be some duplicated transcript assembly, partial gene models, TE genes, or even transcripts from other species such as bacterial contamination. The authors justified the de novo transcriptomic assembly by stating that “genomic data were not available for L. polyactis”. This is also incorrect. A high-quality small yellow croaker is available under NCBI accession number PRJNA705840. Transcriptome data from multiple tissues and developmental stages are also available. The number of protein-coding genes in the small yellow croaker gene is 25,233. The authors must reanalyze the RNA-seq data using the small yellow croaker reference genome JAFMOB000000000.

Minor comment:

     1)      Line 100, “Since genomic data were not available for L. polyactis,” Incorrect. Please cite the little yellow croaker genome paper: Xie Q, Zhan W, Shi Z, Liu F, Niu B, He X, Liu M, Liang Q, Xie Y, Xu P, Wang X, and Lou B (2022). Whole-genome assembly and annotation of little yellow croaker (Larimichthys polyactis) provide insights into hermaphroditism and gonochorism evolution. Authorea. August 02, 2021. DOI: 10.22541/au.162790498.82247747/v1

      2)      Line 68, why did the authors use such an acute noise exposure condition? Any justifications? Does the 30 minutes 120 dB mimic any real-world situation?

      3)      Line 86, in addition to the behavior change, were there any other detrimental consequences to the fish? For example, mortality or other pathological damages?

      4)      Line 88, “illumine” should be “Illumina”

      5)      Line 154, Figure 3, There are three control samples and two in the experimental group. Could the authors explain why?

Author Response

In this manuscript, Zhang et al. investigated the effect of noise exposure on the brain transcriptome. A total of 396 differentially expressed genes were detected, which were enriched in neurological functions. The study is novel and is of interest to the field. However, a technical issue needs to be addressed through reanalysis of the RNA-seq data, before it is suitable for publication at Animals.

Reply: Thanks so much for your suggestion.

We have obtained the gtf annotation files of croaker and reanalyze the RNA-seq date. The GO enrich and KEGG pathway of DEGs have also been performed in clusterProfiler package. New results of RNA-seq were found, we gave it a detailed result and discussion in the revised MS

Major comment:

The authors claim “we assembled and annotated a de novo transcriptome and compared transcript levels in the control group and noise exposure group. Totally 72421 unigenes with high quality.” This is not correct. In general, fish genomes consist of 20 to 40 thousand protein-coding genes. The 72421 genes detected in this research cannot be high-quality. There must be some duplicated transcript assembly, partial gene models, TE genes, or even transcripts from other species such as bacterial contamination. The authors justified the de novo transcriptomic assembly by stating that “genomic data were not available for L. polyactis”. This is also incorrect. A high-quality small yellow croaker is available under NCBI accession number PRJNA705840. Transcriptome data from multiple tissues and developmental stages are also available. The number of protein-coding genes in the small yellow croaker gene is 25,233. The authors must reanalyze the RNA-seq data using the small yellow croaker reference genome JAFMOB000000000.

Reply: Thanks so much for your suggestion.

We have obtained the annotation of little yellow croaker and reanalyze the RNA-seq date. We download the genomoic data from NCBI and perform alignment using hisat2 software. The GO enrich and KEGG pathway of DEGs have also been performed in clusterProfiler package. New DEGs were found, GO and KEGG analysis found more terms and pathway in the DEGs, detailed information can be found in the MS.

Minor comment:

   1)      Line 100, “Since genomic data were not available for L. polyactis,” Incorrect. Please cite the little yellow croaker genome paper: Xie Q, Zhan W, Shi Z, Liu F, Niu B, He X, Liu M, Liang Q, Xie Y, Xu P, Wang X, and Lou B (2022). Whole-genome assembly and annotation of little yellow croaker (Larimichthys polyactis) provide insights into hermaphroditism and gonochorism evolution. Authorea. August 02, 2021. DOI: 10.22541/au.162790498.82247747/v1

Reply: Thanks for your question.

We reanalyzed the RNA-seq dada and also cited this important paper in the MS.

 2)    Line 68, why did the authors use such an acute noise exposure condition? Any justifications? Does the 30 minutes 120 dB mimic any real-world situation?

Reply: Thanks for your suggestions.

The noise exposure of is a not acute condition. The max intensity of a wooden boat noise from field is about 110dB, but owning to the sound reflection of tank wall, the playback noise is up to 120dB in low frequency band (Fig.1). In fact, the intensity of boat noise is positively related to the engine power. The fishing boat we recorded is small and low powered. In the mussel suspended aquaculture farm, some high-powered boat noise may close to or above 120 dB depending on the type of motor, and the boat generally work continuously, more than several hours, when harvesting the mussel.

      3)      Line 86, in addition to the behavior change, were there any other detrimental consequences to the fish? For example, mortality or other pathological damages?

Reply: Thanks for your suggestions.

At the end of the noise exposure, some of fish lost the balance, while the others keep motionless in the water. No fish died after 0.5 h noise exposure. However, in the next day, some fish died. We dissected them and found visceral bleeding in the croaker. The photograph of bleeding is not showed in the MS. Some other studies also reported the bleeding phenomenon in the fish after noise exposure (Halvorsen et al., 2012). In this paper we mainly focused on transcriptomic change in the brain tissues and behavior in croaker after noise exposure, so we do not show the results of visceral bleeding of croaker. In another study related with the swim bladder, we will give the detailed photograph of bleeding. 

Halvorsen M B , Casper B M , Matthews F , et al. Effects of exposure to pile-driving sounds on the lake sturgeon, Nile tilapia and hogchoker[J]. Proceedings of the Royal Society B Biological Sciences, 2012, 279(1748):4705-4714.

      4)      Line 88, “illumine” should be “Illumina”

Reply: Thanks for your suggestions. We have revised it.

    5)      Line 154, Figure 3, There are three control samples and two in the experimental group. Could the authors explain why?

Reply: Thanks for your question.

When we designed the experiment, We prepared three simples for the control and noise expose group. However, the result of RNA-seq found that one brain sample in the noise exposure was polluted, so it was excluded from the further analysis.

Round 2

Reviewer 1 Report

Accepted

Reviewer 2 Report

The authors followed my suggestions and performed the necessary analyses using the Larimichthys polyactis reference genome. The manuscript is much improved, and I recommend the current revision for publication at Animals.